# Upc2-mediated mechanisms of azole resistance in *Candida auris*

Jizhou Li,[1,2] Lola Aubry,[1] Danielle Brandalise,[1] Alix T. Coste,[1] Dominique Sanglard,[1] Frederic Lamoth[1,2]

**ABSTRACT** *Candida auris* is an emerging yeast pathogen of major concern because of its ability to cause hospital outbreaks of invasive candidiasis and to develop resistance to antifungal drugs. A majority of *C. auris* isolates are resistant to fluconazole, an azole drug used for the treatment of invasive candidiasis. Mechanisms of azole resistance are multiple, including mutations in the target gene *ERG11* and activation of the transcription factors Tac1b and Mrr1, which control the drug transporters Cdr1 and Mdr1, respectively. We investigated the role of the transcription factor Upc2, which is known to regulate the ergosterol biosynthesis pathway and azole resistance in other *Candida* spp. Genetic deletion and hyperactivation of Upc2 by epitope tagging in *C. auris* resulted in drastic increases and decreases in susceptibility to azoles, respectively. This effect was conserved in strains with genetic hyperactivation of Tac1b or Mrr1. Reverse transcription PCR analyses showed that Upc2 regulates *ERG11* expression and also activates the Mrr1/Mdr1 pathway. We showed that upregulation of *MDR1* by Upc2 could occur independently from Mrr1. The impact of *UPC2* deletion on *MDR1* expression and azole susceptibility in a hyperactive Mrr1 background was stronger than that of *MRR1* deletion in a hyperactive Upc2 background. While Upc2 hyperactivation resulted in a significant increase in the expression of *TAC1b*, *CDR1* expression remained unchanged. Taken together, our results showed that Upc2 is crucial for azole resistance in *C. auris*, via regulation of the ergosterol biosynthesis pathway and activation of the Mrr1/Mdr1 pathway. Notably, Upc2 is a very potent and direct activator of Mdr1.

**IMPORTANCE** *Candida auris* is a yeast of major medical importance causing nosocomial outbreaks of invasive candidiasis. Its ability to develop resistance to antifungal drugs, in particular to azoles (e.g., fluconazole), is concerning. Understanding the mechanisms of azole resistance in *C. auris* is important and may help in identifying novel antifungal targets. This study shows the key role of the transcription factor Upc2 in azole resistance of *C. auris* and shows that this effect is mediated via different pathways, including the regulation of ergosterol biosynthesis and also the direct upregulation of the drug transporter Mdr1.

**KEYWORDS** ergosterol, efflux pumps, antifungal resistance, zinc cluster transcription factor, *Candida albicans*, *Candida glabrata*

Address correspondence to Frederic Lamoth, Frederic.Lamoth@chuv.ch.

Jizhou Li and Lola Aubry contributed equally to this article. Author order was determined by level of experience and hierarchical position in the laboratory.

F.L. has participated in advisory boards or sponsored symposiums of MSD, Gilead, Pfizer, and Mundipharma and received research grants from Novartis, MSD, Gilead, and Pfizer, outside of the present work. All fees were paid to his institution (CHUV). All other authors declare no conflict of interest.

See the funding table on p. 11.

*C*andida auris is an emerging yeast pathogen that has spread in all continents since 2009 (1). Five genotypically distinct clades have been identified in different geographical areas: Clade I (South Asia), Clade II (East Asia), Clade III (South Africa), Clade IV (South America), and Clade V (Iran) (2). Similarly to other *Candida* spp., *C. auris* can cause severe infections such as candidemia or other forms of invasive candidiasis, accounting now for a substantial and variable proportion (5%–30%) of these infections in some areas (2). Some particular features make *C. auris* a unique and distinct human fungal pathogen, such as its ability to cause nosocomial outbreaks and its potential for the rapid development of resistance to all antifungal drug classes (2). In particular, acquired resistance to fluconazole, an azole drug used for the treatment of invasive

candidiasis, is common in *C. auris*, affecting most isolates (90%–100%) of Clades I and III and variable proportions (15%–60%) of Clades II and IV isolates (2–6).

Mechanisms of azole resistance in *C. auris* are multiple and close to those reported in other *Candida* spp., including mutations in the azole target gene (*ERG11*) and activation of the drug transporters Cdr1 and Mdr1 under the control of their respective transcription factors Tac1b and Mrr1 (7–10). Some gain-of-function (GOF) mutations of *TAC1b* and *MRR1* conferring azole resistance in *C. auris* have been identified (8, 9, 11).

The transcription factor Upc2 is another key regulator of azole resistance in *Candida* spp., which has not yet been investigated in *C. auris* (12, 13). In *Candida albicans* and *Candida glabrata*, Upc2 controls *ERG11* and other genes of the ergosterol biosynthesis pathway (13–17). Moreover, Upc2 was shown to have a possible regulatory function for the drug transporters Cdr1 and Mdr1 (18–21).

The aim of the present study was to assess the role of Upc2 in azole resistance of *C. auris* and to decipher its mechanisms and connections with other pathways of azole resistance.

## RESULTS

### Upc2 is crucial for azole resistance of *C. auris*

To assess the role of Upc2 in the antifungal resistance of *C. auris*, we first deleted *UPC2* in a *C. auris* strain of Clade IV (IV.1) to generate the *upc2Δ* strain (Table 1). Deletion of *UPC2* resulted in a significant decrease in minimal inhibitory concentration (MIC) for both fluconazole and voriconazole compared to the background strain (0.5 vs 4 µg/mL and 0.008 vs 0.03 µg/mL, respectively) (Table 2). We also observed a slight (one dilution) increase in amphotericin B MIC in the *upc2Δ* strain, while susceptibility to micafungin was not affected (Table 2).

We then generated a *C. auris* strain overexpressing *UPC2* (*UPC2^HA^*) using the hyperactivation system for zinc cluster transcription factors by fusion of a 3xHA Tag at its C-terminal locus, as previously described (Table 1) (18, 23). HA tagging of Upc2p in the *UPC2^HA^* strain was confirmed by western blot analysis (Fig. S5). The *UPC2^HA^* strain exhibited higher MICs for both fluconazole and voriconazole compared to the IV.1 strain (32 vs 4 µg/mL and 0.12 vs 0.03 µg/mL, respectively) (Table 2). Of note, we also observed a slight (one dilution) decrease in amphotericin B MIC in the *UPC2^HA^* strain, while micafungin MIC was unchanged (Table 2).

Because overexpression of the drug transporters Cdr1 and Mdr1, under the control of the transcription factors Tac1b and Mrr1, respectively, represents important mechanisms of azole resistance in *C. auris* (8–11), we assessed the impact of *UPC2* deletion in the background of Tac1b or Mrr1 hyperactivation. For this purpose, we generated strains with a C-terminal 3xHA Tag of *TAC1b* or *MRR1* in the wild-type IV.1 strain (*TAC1b^HA^* and *MRR1^HA^*, respectively) and in the *upc2Δ* strain (*TAC1b^HA^/upc2Δ* and *MRR1^HA^/upc2Δ*, respectively; Table 1). Western blot analysis demonstrated HA tagging of Tac1bp and Mrr1p in the *TAC1b^HA^* and *MRR1^HA^* strains, respectively (Fig. S5). While the *TAC1b^HA^* and *MRR1^HA^* strains exhibited fluconazole and voriconazole MICs that were significantly

**TABLE 1** Description of the *C. auris* strains in this study

| Isolate ID | Name | Description | Reference |
|---|---|---|---|
| 17 | IV.1 | Clinical isolate | (22) |
| JLY0133 | *upc2Δ* | IV.1 *upc2Δ::NatR* | This study |
| JLY0095 | *UPC2^HA^* | IV.1 P*_AHD1_*-UPC2-3XHA Tag-T*_ACT1_* ::SAT1 | This study |
| JLY0094 | *TAC1b^HA^* | IV.1 P*_AHD1_*-TAC1b-3XHA Tag-T*_ACT1_* ::SAT1 | This study |
| JLY0051 | *MRR1^HA^* | IV.1 P*_AHD1_*-MRR1-3XHA Tag-T*_ACT1_* ::SAT1 | This study |
| JLY0136 | *TAC1b^HA^/upc2Δ* | IV.1 P*_AHD1_*-TAC1b-3XHA Tag-T*_ACT1_* ::SAT1/upc2Δ::HygR | This study |
| JLY0137 | *MRR1^HA^/upc2Δ* | IV.1 P*_AHD1_*-MRR1-3XHA Tag-T*_ACT1_* ::SAT1/upc2Δ::HygR | This study |
| JLY0129 | *UPC2^HA^/mdr1Δ* | IV.1 P*_AHD1_*-UPC2-3XHA Tag-T*_ACT1_* ::SAT1/mdr1Δ::HygR | This study |
| JLY0130 | *UPC2^HA^/mrr1Δ* | IV.1 P*_AHD1_*-UPC2-3XHA Tag-T*_ACT1_* ::SAT1/mrr1Δ::HygR | This study |

**TABLE 2** MIC values to antifungal drugs of the different strains of this study

| Strain | Fluconazole MIC (µg/mL) | Voriconazole MIC (µg/mL) | Amphotericin B MIC (µg/mL) | Micafungin MIC (µg/mL) |
|---|---|---|---|---|
| IV.1 | 4 | 0.03 | 2 | 0.125 |
| *upc2Δ* | 0.5 | 0.008 | 4 | 0.125 |
| *UPC2^HA* | 32 | 0.125 | 1 | 0.125 |
| *TAC1b^HA* | 128 | 2 | 2 | 0.125 |
| *TAC1b^HA/upc2Δ* | 2 | 0.015 | 4 | 0.125 |
| *MRR1^HA* | 32 | 0.125 | 2 | 0.125 |
| *MRR1^HA/upc2Δ* | 1 | 0.015 | 4 | 0.125 |
| *UPC2^HA/mdr1Δ* | 32 | 0.125 | 1 | 0.125 |
| *UPC2^HA/mrr1Δ* | 16 | 0.06 | 1 | 0.125 |

higher compared to the IV.1 strain (i.e., considered fluconazole-resistant according to the tentative MIC breakpoints) (24), *UPC2* deletion could restore azole susceptibility in the *TAC1b^HA/upc2Δ* and *MRR1^HA/upc2Δ* strains (Table 2).

These results show that Upc2 is required for azole resistance of *C. auris*.

As a next step, we investigated the mechanisms by which Upc2 controls azole resistance in *C. auris* and its connections with other pathways of azole resistance, such as the ergosterol biosynthesis pathway and the drug transporter systems Tac1b/Cdr1 and Mrr1/Mdr1. For this purpose, we performed real-time reverse transcription PCR (RT-PCR) analyses to measure *UPC2*, *ERG11*, *TAC1b*, *CDR1*, *MRR1*, and *MDR1* expressions in the different mutant strains of this study.

## Upc2 is independent of Tac1b and Mrr1

We first performed RT-PCR analyses of *UPC2* expression (Fig. 1A). As expected, we observed a complete loss of *UPC2* expression in the *upc2Δ* strain and a significant overexpression of *UPC2* (18.61-fold) in the *UPC2^HA* strain compared to the IV.1 strain. *UPC2* expression was not affected by hyperactivation of Tac1b or Mrr1 in the *TAC1b^HA* and *MRR1^HA* strains, respectively. These results show that Upc2 is not regulated by Tac1b or Mrr1.

## Upc2 regulates Erg11

Because Upc2 is known to control the ergosterol biosynthesis pathway via regulation of Erg11 in other *Candida* spp. (13, 15, 16, 18), we also measured *ERG11* expression (Fig. 1B). As expected, *ERG11* transcript levels were significantly decreased in the *upc2Δ* strain and increased in the *UPC2^HA* strain compared to the IV.1 strain. Hyperactivation of Tac1b induced a significant decrease in *ERG11* expression, which was not the case following Mrr1 hyperactivation.

These results confirm that Upc2 controls azole resistance of *C. auris* via regulation of Erg11. We also observed that upregulation of Tac1b was associated with downregulation of the ergosterol biosynthesis pathway, possibly as a compensatory effect.

## Upc2 induces azole resistance independently of Cdr1

Because previous studies in *C. glabrata* suggest a link between Upc2, the transcription factor Pdr1 (ortholog of *C. auris* Tac1b), and its target the drug transporter Cdr1 (13, 21), we then measured *TAC1b* and *CDR1* expressions in the different generated strains (Fig. 2A and B). While deletion of *UPC2* did not result in significant changes in *TAC1b* and *CDR1* expressions, we observed a significant increase in *TAC1b* expression but not *CDR1*, in the *UPC2^HA* strain. As expected, the *TAC1b^HA* strain displayed strong overexpression of both *TAC1b* and *CDR1*, which could not be reduced by *UPC2* deletion. On the contrary, we observed a slight overexpression of *TAC1b* in the *TAC1b^HA/upc2Δ* compared to the *TAC1b^HA* strain, which may correspond to some compensatory effect.

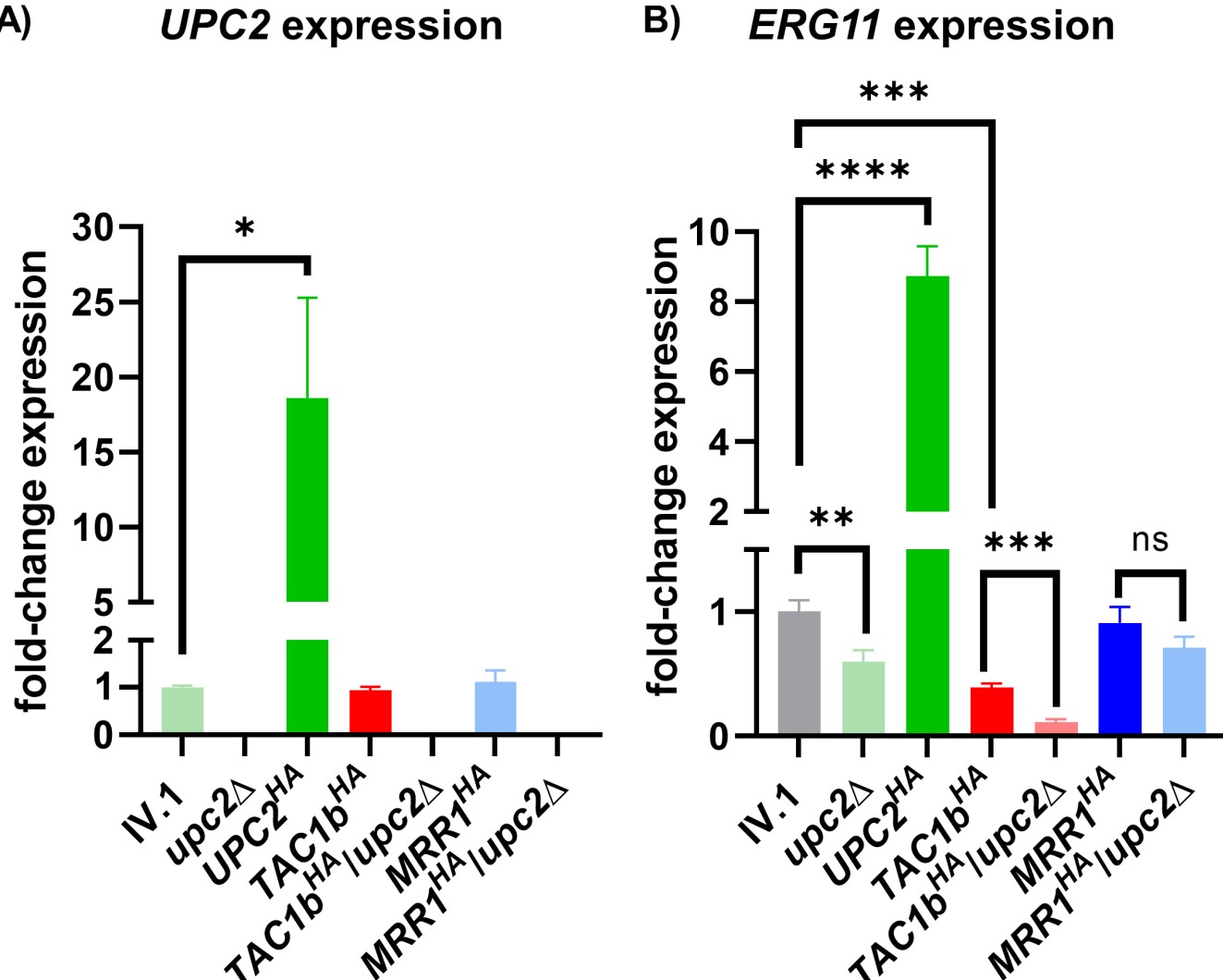

**FIG 1** Relative expression of *UPC2* (A) and *ERG11* (B) in the different strains of this study. Results are expressed as fold-change compared to the wild-type IV.1 strain. Bars represent means with standard deviations of three biological replicates. The brackets show the conditions that were compared using the *t*-test method and interpreted as statistically significant: * *P*-value ≤ 0.05, ** *P*-value ≤ 0.01, *** *P*-value ≤ 0.001, and **** *P*-value ≤ 0.0001; not statistically significant: ns.

Expression of *TAC1b* and *CDR1* was unchanged in the *MRR1^HA* and *MRR1^HA/upc2Δ* strains compared to the IV.1 strain, which suggests that the Tac1b/Cdr1 pathway is independent of Mrr1.

Taken together, these results show that Upc2 activation does not significantly impact *CDR1* expression despite some upregulation of *TAC1b*.

## Upc2 regulates Mrr1 and Mdr1

Because previous works in *C. albicans* have suggested a possible link between Upc2 and the Mrr1/Mdr1 pathway (14, 18), we then performed RT-PCR analyses of *MRR1* and *MDR1* expressions in the different mutant strains (Fig. 3A and B). No significant changes in *MDR1* expression were found in the *upc2Δ* strain compared to the IV.1 strain. However, we observed a significant increase in *MRR1* and *MDR1* expressions in the *UPC2^HA* strain. Notably, *MDR1* expression resulting from Upc2 hyperactivation was particularly high (>60-fold compared to that of the wild-type IV.1 strain). As expected, hyperactivation of Mrr1 in the IV.1 strain resulted in significant upregulation of both *MRR1* and *MDR1*. Of note, expression of *MDR1* was significantly higher in the *UPC2^HA* strain compared to the

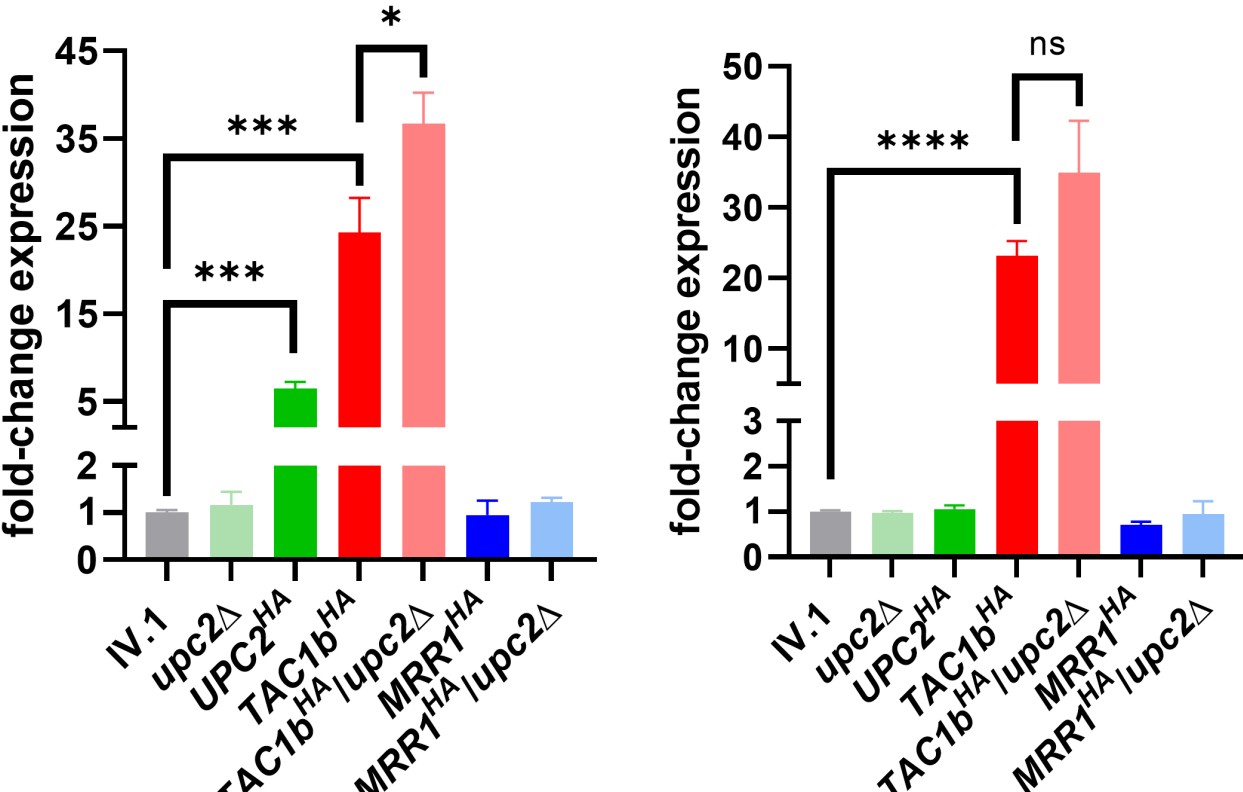

**FIG 2** Relative expression of *TAC1b* (A) and *CDR1* (B) in the different strains of this study. Results are expressed as fold-change compared to the wild-type IV.1 strain. Bars represent means with standard deviations of three biological replicates. The brackets show the conditions that were compared using the *t*-test method and interpreted as statistically significant: * *P*-value ≤ 0.05, ** *P*-value ≤ 0.01, *** *P*-value ≤ 0.001, and **** *P*-value ≤ 0.0001; not statistically significant: ns.

*MRR1*<sup>HA</sup> strain. *UPC2* deletion in the *MRR1*<sup>HA</sup> background resulted in a significant decrease of both *MRR1* and *MDR1* expressions.

No significant changes in *MRR1* and *MDR1* expressions were observed in the *TAC1b*<sup>HA</sup> and *TAC1b*<sup>HA</sup>/*upc2*Δ strains compared to the IV.1 strain, which further suggests that the Tac1b/Cdr1 and Mrr1/Mdr1 pathways are independent of each other. Globally, these results show that Upc2 can upregulate both Mrr1 and Mdr1.

### Upc2 can control Mdr1 via mechanisms that are independent of Mrr1

To further investigate the link between Upc2 and the Mrr1/Mdr1 pathway, we generated two strains with deletions of *MRR1* and *MDR1* in the *UPC2*<sup>HA</sup> strain (*UPC2*<sup>HA</sup>/*mrr1*Δ and *UPC2*<sup>HA</sup>/*mdr1*Δ, respectively; Table 1). Fluconazole and voriconazole MICs were slightly decreased in the *UPC2*<sup>HA</sup>/*mrr1*Δ, but not in the *UPC2*<sup>HA</sup>/*mdr1*Δ strain, when compared to the *UPC2*<sup>HA</sup> strain (Table 2). RT-PCR analyses showed that *MDR1* deletion had no impact on *MRR1* expression in the *UPC2*<sup>HA</sup> strain background (Fig. 3A). Surprisingly, high level of *MDR1* expression could be maintained in the Upc2-hyperactivated strain despite the loss of *MRR1* (Fig. 3B). *MDR1* expression in the *UPC2*<sup>HA</sup>/*mrr1*Δ strain was only slightly decreased compared to the *UPC2*<sup>HA</sup> strain and similar to that observed in the *MRR1*<sup>HA</sup> strain. On the contrary, a significant decrease in *MDR1* expression was observed with the loss of *UPC2* in the *MRR1*<sup>HA</sup> background.

These results demonstrate that Upc2 can directly regulate Mdr1 independently from Mrr1. Moreover, the role of Upc2 in controlling Mdr1 appears to be stronger than that of

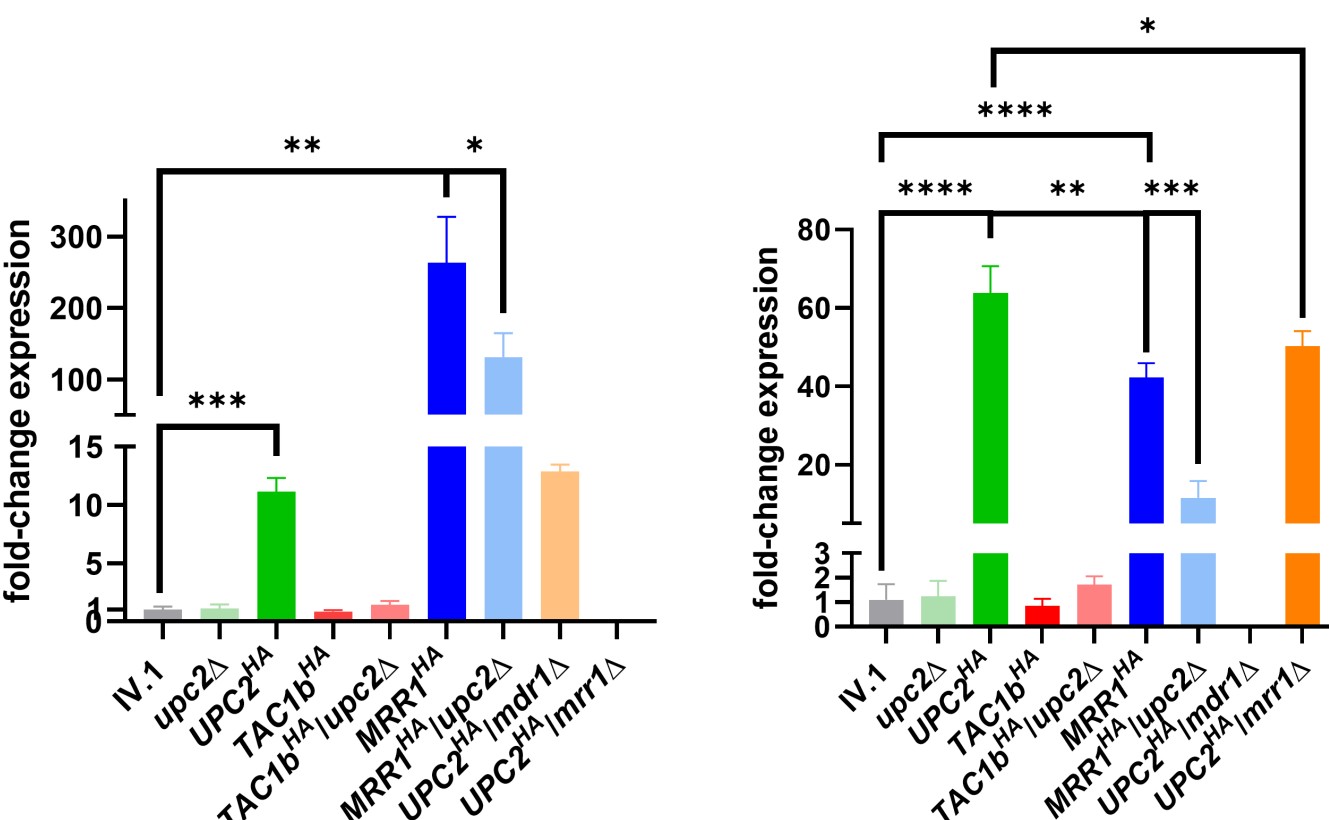

**FIG 3** Relative expression of *MRR1* (A) and *MDR1* (B) in the different strains of this study. Results are expressed as fold-change compared to the wild-type IV.1 strain. Bars represent means with standard deviations of three biological replicates. The brackets show the conditions that were compared using the *t*-test method and interpreted as statistically significant: * *P*-value ≤ 0.05, ** *P*-value ≤ 0.01, *** *P*-value ≤ 0.001, and **** *P*-value ≤ 0.0001.

Mrr1. However, hyperactivated Upc2 can maintain azole resistance even in the absence of Mdr1, probably via its effect on the ergosterol biosynthesis pathway.

## DISCUSSION

The transcription factor Upc2 was shown to be a key regulator of azole resistance in *Candida* spp. (12, 20). In this study, we analyzed the Upc2-dependent pathways of azole resistance in *C. auris*. Our results, summarized in the schematic representation of Fig. 4, show some similar and distinct mechanisms of Upc2-mediated azole resistance in *C. auris* compared to *C. albicans* or *C. glabrata*.

First, Upc2 is known to regulate Erg11 and the ergosterol biosynthesis pathway in *C. albicans* and *C. glabrata* (12, 13, 15, 16, 18). In the present study, we demonstrated that Upc2 controls the same pathway in *C. auris*. Deletion of *UPC2* resulted in decreased *ERG11* expression and reduced resistance to fluconazole and voriconazole, while its hyperactivation led to the opposite effect.

Previous analyses in *C. albicans* showed that Upc2 regulates other genes involved in azole resistance, such as drug transporters of the major facilitator superfamily (MFS) or ATP-binding cassette (ABC) family (14, 17). Two of these drug transporters, Cdr1 (ABC) and Mdr1 (MFS), which are mainly controlled by the transcription factors Tac1 and Mrr1, respectively, are key regulators of azole resistance in *Candida* spp. (25, 26). Genome-wide location profiling analyses found that Upc2 binds to the promoters of *MDR1* and *CDR1*

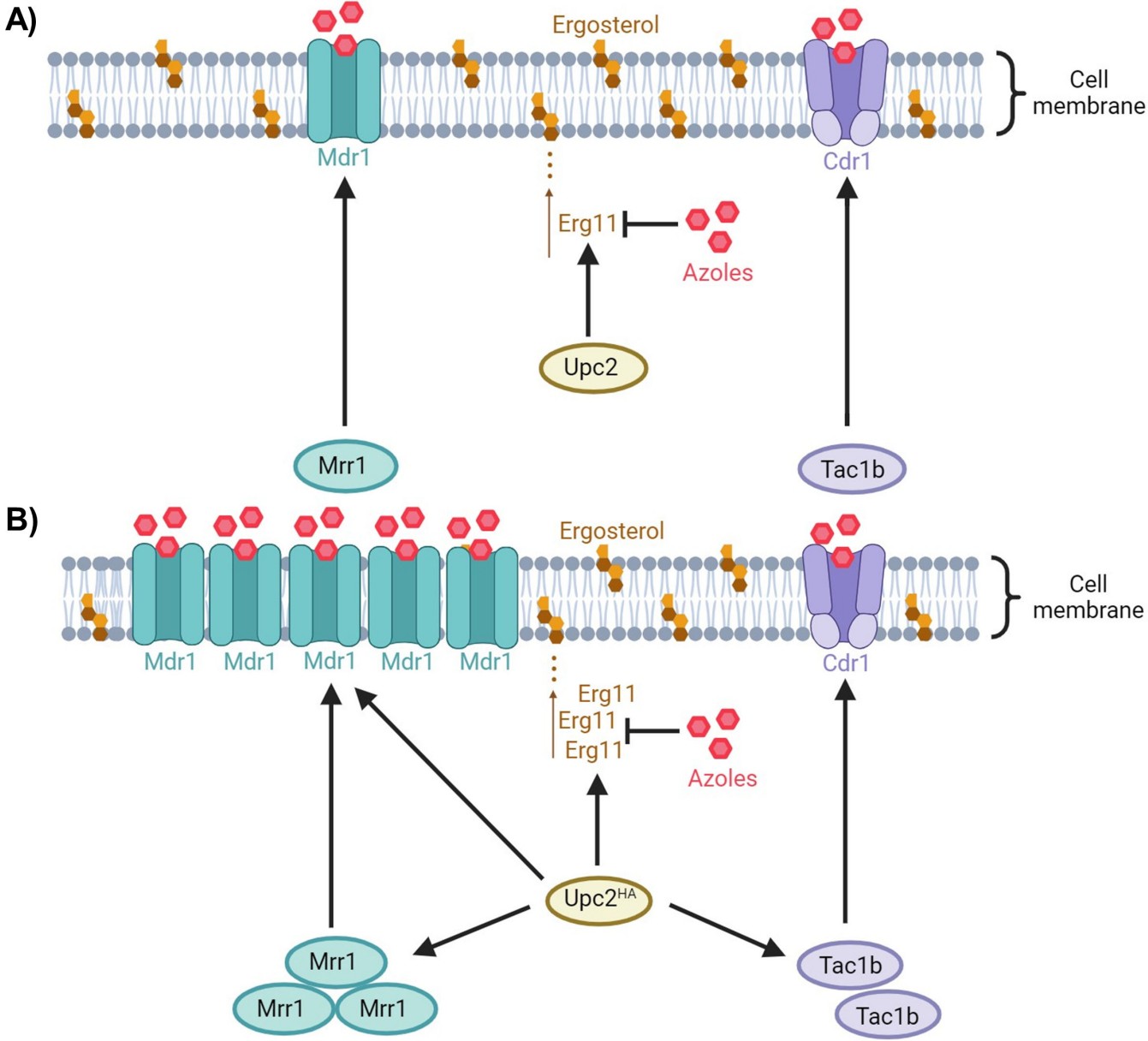

**FIG 4** Schematic representation of the role of Upc2 in azole resistance of *C. auris*. In standard conditions (A), Upc2 regulates Erg11 (target of the azole drugs). Upc2 hyperactivation [Upc2^HA (B)] results in upregulation of Erg11. Moreover, Upc2 induces high level of Mdr1 by direct upregulation and indirect control via Mrr1. Although Upc2 hyperactivation results in some upregulation of Tac1b, Cdr1 levels remain unchanged. Black arrows = positive regulation.

in *C. albicans* (17). However, the actual role of Upc2 in the regulation of these genes is unclear. Some GOF mutations in *UPC2* were found to be associated with increased *MDR1* expression in *C. albicans* (14). However, Mdr1 upregulation via Mrr1 hyperactivation or chemical compounds occurred independently of Upc2, while activation of Upc2 only resulted in a slight upregulation of Mdr1, which was dependent on Mrr1 (18). Among ABC transporters, only Cdr11, but not Cdr1, was found to be activated by Upc2 in *C. albicans* (14). In *C. glabrata*, deletion of *UPC2A* (*UPC2* ortholog) in a strain harboring a GOF mutation of *PDR1* (*TAC1* ortholog) could drastically reduce azole resistance despite conserved high expression levels of *CDR1* (13). However, its deletion in an azole-suscepti-ble strain could partially reduce the fluconazole-induced expression of both *PDR1* and *CDR1* (13). Indeed, other studies showed that *UPC2A* could directly bind to the *PDR1* and

*CDR1* promoters (19, 21). Interestingly, inhibition of the ergosterol biosynthesis pathway by different approaches led to increased expression of *PDR1* and *CDR1*, which could be reduced following *UPC2* deletion (21). Transcriptomic analyses in *C. glabrata* showed that Upc2 could induce expression of some membrane protein genes that are also upregulated by Pdr1, which further supports the link between Upc2 and the Pdr1/Cdr1 pathway in this yeast pathogen (20).

We investigated the link between Upc2 and the Tac1b/Cdr1 and Mrr1/Mdr1 pathways in *C. auris*. We found that strong hyperactivation of Tac1b or Mrr1 was not sufficient to maintain azole resistance in the absence of Upc2. These results are consistent with previous observations in *C. albicans* and *C. glabrata* (12, 13). We observed that hyperactivation of Upc2 resulted in a significant overexpression of *TAC1b*, which was, however, much inferior to that obtained in the Tac1b-hyperactivated strain and possibly therefore not sufficient to induce an increase in *CDR1* expression. We concluded that Cdr1 upregulation was not sufficient to induce azole resistance in the absence of Upc2, as previously observed in *C. glabrata* (13). Our results also support a possible regulation of Tac1b by Upc2 in *C. auris*, which has been demonstrated in *C. glabrata* (21). However, this effect seems marginal in *C. auris*, as it did not result in any impact on *CDR1* expression, despite high overexpression of *UPC2*.

The link between Upc2 and the Mrr1/Mdr1 pathway seems more relevant in *C. auris* according to our results. Indeed, we observed that Upc2 hyperactivation could induce significant overexpression of both *MRR1* and *MDR1*. Interestingly, Upc2 hyperactivation reached a level of *MRR1* expression that was lower compared to that observed in the Mrr1-hyperactivated strain but level of *MDR1* expression that was higher. This suggested that Upc2 can control Mdr1 by mechanisms that are mediated by Mrr1 but also independent of Mrr1. This was further confirmed by demonstrating the ability of Upc2 to induce overexpression of *MDR1* even in the absence of Mrr1. *MDR1* expression was actually more impacted by *UPC2* deletion in the hyperactive Mrr1 background than by *MRR1* deletion in the hyperactive Upc2 background, which suggests that Upc2 may be a predominant transcription factor controlling Mdr1. These results differ from those previously reported in *C. albicans*, where the impact of Upc2 activation on *MDR1* expression was modest and remained dependent on Mrr1 (18). The regulatory mechanisms behind the control of *MDR1* by *UPC2* remain unexplored, and future studies should address the occupancy of Upc2 in the *MDR1* promoter of *C. auris*. Moreover, the actual relevance of this mechanism of azole resistance mediated by Mdr1 via Upc2 remains unclear. Indeed, we also observed that hyperactivation of Upc2 could maintain high level of azole resistance even in the absence of Mdr1, which suggests that the major impact of Upc2 on azole resistance is probably mediated via upregulation of Erg11.

Interestingly, our data also show that *UPC2* deletion or hyperactivation had distinct effects on amphotericin B MIC (i.e., slight MIC increase and decrease, respectively), which may result from the impact of Upc2 on the ergosterol biosynthesis pathway.

In conclusion, this study shows that Upc2 is a major and universal regulator of azole resistance in *C. auris*, cross-talking with other important resistance pathways, as it has been previously demonstrated in other *Candida* spp. However, our results highlighted distinct features about the connections of Upc2 with other pathways of azole resistance in *C. auris* compared to *C. albicans* and *C. glabrata*. Indeed, in *C. auris*, the link between Upc2 and the Mrr1/Mdr1 pathway seems predominant when compared to the two other species, while the link with the Tac1b/Cdr1 pathway appears to be less relevant than that observed in *C. glabrata*.

GOF mutations in transcription factors have often been linked to drug resistance of fungal pathogens (27). In *C. auris*, such mutations have been identified in Tac1b and Mrr1 (8, 9, 11). Whether acquired GOF mutations of Upc2 may be responsible for azole resistance in *C. auris* clinical isolates is still unknown. Further analyses of clinical isolates are warranted to assess the actual relevance of Upc2-mediated azole resistance mechanisms in *C. auris*.

## MATERIALS AND METHODS

### Strains, plasmids, and media

The *C. auris* strains and the plasmids used in this study are listed in Table 1 and Table S1, respectively. Plasmids pJK795 containing the *NatR* cassette and pYM70 containing the *HygR* cassette were used as sources of nourseothricin and hygromycin resistances, respectively (28, 29). The plasmid Clp-p*ACT1*-3xFLAG-MNase-SV40-*CYC-SAT1* (a gift from Adnane Sellam, Institute of Cardiology of Montreal, Canada), which contains the nourseothricin resistance cassette *SAT1* and *C. auris* neutral site *CauNI*, was used for constructs of the hyperactivated zinc cluster transcription factors (30). Isolates were grown in yeast extract-peptone-dextrose (YEPD) medium. Cultures were incubated for 16–20 h at 37°C on solid YEPD agar plates or in liquid YEPD under constant agitation (220 rpm).

The antifungal drugs fluconazole, voriconazole, amphotericin B (Sigma-Aldrich, St. Louis, MO), and micafungin (Selleck Chemicals, Houston, TX) were obtained as powder suspensions and dissolved in dimethyl sulfoxide (DMSO) as stocks of 10 mg/mL for fluconazole and 1 mg/mL for other drugs.

*Escherichia coli* DH5α was used as a host for plasmid constructions and propagation. DH5α was grown in Luria-Bertani broth or agar plates supplemented with ampicillin (100 μg/mL, AppliChem, Darmstadt, Germany) when required for 16–20 h at 37°C. Plasmids were extracted with the Plasmid Mini Kit (Qiagen, Hilden, Germany). All primers used in this study are listed in Table S2. All DNA sequences used for plasmid constructions were amplified from the genome of *C. auris* strain IV.1 unless otherwise specified.

### Construction of plasmids for hyperactivation of zinc finger transcription factors

For hyperactivation of zinc cluster transcription factors, we used a model where the target gene is put under the control of the *ADH1* promoter and tagged at its C-terminal locus by a 3xHA sequence, as previously described (Fig. S1) (18, 23). The *ADH1* promotor was amplified by the primers ADH1p_PF_KpnI and ADH1p_PR_KasI_NheI. The PCR product digested by KpnI and NheI was cloned at the NheI/KpnI sites of the plasmid Clp-p*ACT1*-3xFLAG-MNase-SV40-*CYC-SAT1* to generate pjli6.

For the hyperactivation of Mrr1, the *MRR1* nucleotide sequence was amplified by the primers MRR1_PF_KasI and MRR1_PR_BsrGI_Tag_CS_NruI_NheI (containing the 3xHA Tag sequence). The PCR product was digested by the primers KasI and NheI and cloned at KasI/NheI sites of pjli6 to generate pjli7. The *ACT1* terminator was amplified by the primers ACT1t_PF_NheI and ACT1t_PR_NheI, and the PCR product was then cloned at NheI site of pjli7 to generate pjli8.

For the hyperactivation of Tac1b, the nucleotide sequence of *TAC1b* was amplified by the primers TAC1b_PF_KasI and TAC1b_PR_BsrGI. The PCR product was digested by KasI and BsrGI and then cloned at KasI/BsrGI sites of pjli8 to generate pjli11.

For the hyperactivation of Upc2, the nucleotide sequence of *UPC2* was amplified by the primers UPC2_PF_KasI and UPC2_PR_BsrGI. The PCR product was digested by KasI and BsrGI and then cloned at KasI/BsrGI sites of pjli8 to generate pjli12. Absence of mutations resulting from PCR amplification was verified by sequencing. Plasmids pjli8, pjli11, and pjli12 were linearized by StuI for integration in the *CauNI* locus of the IV.1 strain to generate the strains *MRR1*[HA], *TAC1b*[HA], and *UPC2*[HA] (for hyperactivation of Mrr1, Tac1b, and Upc2, respectively).

### Constructs for *UPC2* deletion

For *UPC2* deletion in the IV.1 strain, the construct was made by fusion PCR of three PCR products (Fig. S2A). The first PCR product consisted of the 5′ flanking region (580 bp) of *UPC2* and was amplified with primers UPC2_del_PF1 and UPC2_del_PR1. The second PCR product consisted of the *NatR* cassette, which was amplified from pJK795 with

primers UPC2_del_PF2 and UPC2_del_PR2. The third PCR product consisted of the 3′ flanking region (520 bp) of *UPC2* and was amplified with primers UPC2_del_PF3 and UPC2_del_PR3. The three PCR products were purified with the QIAquick PCR Purification Kit (Qiagen, Hilden, Germany). The fusion PCR was performed with the nested primers UPC2_del_PF4 and UPC2_del_PR4 in the presence of 1.3 M of betaine.

For *UPC2* deletion in the *TAC1b^HA* and *MRR1^HA* strains, the selection marker *HygR* (hygromycin resistance) was used in place of *NatR* (nourseothricin resistance), which was already used for the generation of these mutant strains (Fig. S2B). The same flanking regions of *UPC2* were amplified using primers UPC2_del_PF1 and UPC2_del_PR1(hygro) for the 5′ flanking region and UPC2_del_PF3(hygro) and UPC2_del_PR3 for the 3′ flanking region. The *HygR* gene was amplified from the plasmid pYM70 with primers UPC2_del_PF2(hygro) and UPC2_del_PR2(hygro). The fusion PCR was performed with the nested primers UPC2_del_PF4 and UPC2_del_PR4.

The fusion PCR products were used for transformation in IV.1, *TAC1b^HA*, and *MRR1^HA* strains to generate strains *upc2Δ*, *TAC1b^HA/upc2Δ*, and *MRR1^HA/upc2Δ*, respectively (Fig. S3).

## Constructs for *MRR1* and *MDR1* deletion in the *UPC2^HA* strain

Construction of *MDR1* and *MRR1* deletion cassettes was performed as described in our previous publication, using the *HygR* cassette as a selection marker to substitute the target gene (8). The fusion PCR products were used for transformation in strain *UPC2^HA* to generate strains *UPC2^HA/mrr1Δ* and *UPC2^HA/mdr1Δ*, with deletion of *MRR1* and *MDR1*, respectively.

## Transformations in *C. auris*

Transformations in *C. auris* were performed by a CRISPR-Cas9 approach. RNA-protein complexes (RNPs) reconstituted with purified Cas9 protein combined with scaffold- and gene-specific guide RNAs were used as previously described (31). Gene-specific RNA guides were designed to contain 20 bp homologous sequences of the upstream and downstream regions of the target region for integration. Sequences of RNA guides are shown in Table S2. The mix of the guide RNAs, the Cas9 endonuclease 3NLS (Integrated DNA Technologies Inc., Coralville, IA), and tracrRNA (universal transactivating CRISPR RNA) were prepared according to a previously described protocol (32).

Transformation of *C. auris* was performed by electroporation with about 1 μg of the constructs as previously described (32). Transformants were selected at 37°C on YEPD containing 200 μg/mL of nourseothricin (Werner BioAgents, Jena, Germany) for transformations in the IV.1 strain or 600 μg/mL of hygromycin B (Corning, Corning, NY) for transformations in the *TAC1b^HA* and *MRR1^HA* strains. Correct integration of the constructs was verified by PCR screening for all the HA-tagged strains (Fig. S4A and B), the *UPC2* deletion strains (Fig. S6A through C), the *MDR1* deletion strains (Fig. S7A and B), and the *MRR1* deletion strains (Fig. S8A and B).

## Protein extraction and western blot analysis

Protein extracts were obtained from whole cell extracts of the IV.1, *MRR1^HA*, *TAC1b^HA*, and *UPC2^HA* strains after precipitation with trichloroacetic acid as previously described (33). Proteins were separated through electrophoresis using Mini-PROTEAN TGX gel (Bio-Rad Laboratories, Hercules, CA) and blotted onto nitrocellulose membranes. Mrr1p, Tac1bp, and Upc2p were detected by primary HA Tag Monoclonal Antibody (Invitrogen, ThermoFisher Scientific, Waltham, MA) with 1/2,500 dilution in 5% bovine serum albumin in phosphate-buffered saline with 1% Tween (BSA-PBS-T) and secondary Goat anti-Mouse IgG (H + L) antibody (Invitrogen, ThermoFisher Scientific, Waltham, MA) with 1/500 dilution in BSA-PBS-T.

## Antifungal susceptibility testing

MIC of fluconazole, voriconazole, amphotericin B, and micafungin were determined for the *C. auris* isolates according to the procedure of the Clinical and Laboratory Standards Institute (CLSI, M27, 4th edition) (34).

## RT-PCR

Each *C. auris* isolate was grown overnight in 5 mL of liquid YEPD under constant agitation at 37°C. Cultures were diluted to a density of $0.75 \times 10^7$ cells/mL in 5 mL of fresh YEPD and were grown at 37°C under constant agitation for 2–3 h to reach a density of $1.5 \times 10^7$ cells/mL. Total RNA was extracted with the Quick-RNA Fungal/Bacterial Miniprep Kit (Zymo Research, Freiburg im Breisgau, Germany), and total RNA extracts were treated with DNase using the DNA-Free Kit (Thermo Fisher Scientific Inc., Waltham, MA). The concentration of purified RNA was measured with a NanoDrop 1000 instrument (Witec AG, Sursee, Switzerland). RNA was stored at −80°C until use. Each isolate was prepared in biological triplicates.

One microgram of RNA of each isolate was converted into cDNA using the Transcriptor high-fidelity cDNA synthesis kit (Roche, Basel, Switzerland). RT-PCR was performed in 96-well plates using the PowerUp SYBR Green Master Mix (Applied Biosystems, Waltham, MA) with primers of the targeted genes (Supplementary Material S1) and supplemented with nuclease-free water up to 20 µL for each reaction. The primers used for *ACT1*, *CDR1*, *MDR1*, *ERG11*, *TAC1b*, *MRR1*, and *UPC2* amplification were mentioned in Table S2. Each experiment was performed in biological triplicates and technical duplicates. The QuantStudio Software Program (Thermo Fisher Scientific Inc., Waltham, MA) including a melt curve stage was used for real-time PCR with activation at 95°C for 10 min, 40 cycles of denaturation at 95°C for 15 s, and annealing/extension at 60°C for 1 min. Gene expression was calculated with the threshold cycle ($2^{-\Delta\Delta CT}$) method and normalized to the *ACT1* expression (35). Results were analyzed by the *t*-test method (GraphPad).

## ACKNOWLEDGMENTS

This study was supported by grants from the Swiss National Science Foundation (SNSF, project number 310030_192611) and the Santos-Suarez Foundation.

## AUTHOR AFFILIATIONS

[1]Department of Laboratory Medicine and Pathology, Institute of Microbiology, Lausanne University Hospital, University of Lausanne, Lausanne, Switzerland
[2]Infectious Diseases Service, Department of Medicine, Lausanne University Hospital, University of Lausanne, Lausanne, Switzerland

## AUTHOR ORCIDs

Jizhou Li http://orcid.org/0000-0001-6007-2661
Alix T. Coste http://orcid.org/0000-0001-9481-9778
Dominique Sanglard http://orcid.org/0000-0002-5244-4178
Frederic Lamoth http://orcid.org/0000-0002-1023-5597

## FUNDING

| Funder | Grant(s) | Author(s) |
| --- | --- | --- |
| Swiss National Science Foundation | 310030_192611 | Frederic Lamoth |
| Fondation Santos-Suarez pour la Recherche Médicale (Santos-Suarez Foundation for Medical Research) | | Frederic Lamoth |

## AUTHOR CONTRIBUTIONS

Jizhou Li, Conceptualization, Data curation, Formal analysis, Investigation, Methodology, Writing – original draft | Lola Aubry, Conceptualization, Data curation, Formal analysis, Investigation, Methodology | Danielle Brandalise, Data curation, Formal analysis, Investigation | Alix T. Coste, Conceptualization, Methodology, Writing – review and editing | Dominique Sanglard, Conceptualization, Formal analysis, Investigation, Methodology, Supervision, Writing – review and editing | Frederic Lamoth, Conceptualization, Formal analysis, Funding acquisition, Methodology, Project administration, Resources, Supervision, Writing – original draft

## ADDITIONAL FILES

The following material is available online.

### Supplemental Material

**Supplemental material (Spectrum03526-23-s0001.pdf).** Tables S1 and S2; Figures S1 to S8.

### Open Peer Review

**PEER REVIEW HISTORY (review-history.pdf).** An accounting of the reviewer comments and feedback.

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
