## [Reviewer comments · Microbiology Spectrum]

Microbiology Spectrum

Upc2-mediated mechanisms of azole resistance in *Candida auris*

Jizhou Li, Lola Aubry, Danielle Brandalise, Alix Coste, Dominique Sanglard, and Frederic Lamoth

Corresponding Author(s): Frederic Lamoth, Centre Hospitalier Universitaire Vaudois

Review Timeline:

Submission Date:	October 3, 2023
Editorial Decision:	November 7, 2023
Revision Received:	December 12, 2023
Accepted:	December 13, 2023

Editor: Dimitrios Kontoyiannis

Reviewer(s): Disclosure of reviewer identity is with reference to reviewer comments included in decision letter(s). The following individuals involved in review of your submission have agreed to reveal their identity: Kelly Ishida (Reviewer #2)

Transaction Report:

DOI: <https://doi.org/10.1128/spectrum.03526-23>

Re: Spectrum03526-23 (Upc2-mediated mechanisms of azole resistance in *Candida auris*)

Dear Prof. Frederic Lamoth:

Thank you for the privilege of reviewing your work. Below you will find my comments, instructions from the Spectrum editorial office, and the reviewer comments.

Revision Guidelines

Sincerely,
Dimitrios Kontoyiannis
Editor
Microbiology Spectrum

Reviewer #1 (Comments for the Author):

This is a strong manuscript that characterizes the role of Upc2 in *C. auris* azole susceptibility. The data strongly support the conclusions and add interesting observations that make important extensions to what has been observed for the homologous TF in other *Candida* spp. I have only a few suggestions that could improve the manuscript.

1. It would be nice to use the HA tag of the hyperactive Upc2 to confirm that protein levels are also increased.
2. The statistical tests need to be corrected for multiple comparisons. I do not think this will affect the results but it needs to be

corrected to maintain rigor.

3. A figure to summarize the interactions of Upc2 might make the discussion a little more straightforward for the reader, particularly those not familiar with the azole field.

4. Line 42. Azoles are not a first line therapy for any candida spp. causing invasive disease according to IDSA and European guidelines. This is because of the likelihood of resistance and the fact that echinocandins are more effective clinically.

Reviewer #2 (Comments for the Author):

The manuscript provides interesting and meaningful results about the role of transcription factor Upc2 on azole resistance mechanisms in *Candida auris*, that 90-100 clinical isolates are fluconazole resistant.

The manuscript is well written and easy to read and interpret the results.

Minor comments:

insert the statistical test used in the captions.

In the figures, try changing the arrows by other symbols that indicate the differences between the compared strains. It's confusing.

MS Spectrum 03526-23: responses to reviewers

Reviewer #1 (Comments for the Author):

This is a strong manuscript that characterizes the role of Upc2 in *C. auris* azole susceptibility. The data strongly support the conclusions and add interesting observations that make important extensions to what has been observed for the homologous TF in other *Candida* spp. I have only a few suggestions that could improve the manuscript.

1. It would be nice to use the HA tag of the hyperactive Upc2 to confirm that protein levels are also increased.

Response: We fully agree that demonstration of the hyperactivation of Upc2 at the protein level would strengthen the manuscript. We have performed a Western blot, which indicates the protein levels of Upc2, Mrr1 and Tac1 in the UPC2^{HA}, MRR1^{HA} and TAC1b^{HA} strains, respectively. See additional text in the results and methods section (lines 81-82 and 341-350) and new Supplementary Figure 5.

2. The statistical tests need to be corrected for multiple comparisons. I do not think this will affect the results but it needs to be corrected to maintain rigor.

Response: We are sorry, but we do not understand the request of the reviewer. We have performed RT-PCR experiments in biological triplicates and technical duplicates, as mentioned in the method section (lines 372). Comparison were performed using the t test (lines 377). We have experience with RT-PCR and we have always published our results using this statistical approach (see our previous publications). This is also the standard approach used in other publications reporting RT-PCR experiments. To our knowledge, tests for multiple comparisons are applied when multiple variables are tested on a same sampling, which is not the case here. Results in the Figures 1, 2 and 3 show individual comparisons for only two conditions. This has been clarified in the figure legends

3. A figure to summarize the interactions of Upc2 might make the discussion a little more straightforward for the reader, particularly those not familiar with the azole field.

Response: we agree with this proposition. We have added a Figure 4 with a schematic representation of Upc2 function in *Candida auris*. We have added a sentence at the beginning of the discussion section to introduce this new figure (lines 167-170).

4. Line 42. Azoles are not a first line therapy for any *Candida* spp. causing invasive disease according to IDSA and European guidelines. This is because of the likelihood of resistance and the fact that echinocandins are more effective clinically.

Response: We agree and we have modified this sentence in the abstract and in the introduction (line 5-7 and 42-45).

Reviewer #2 (Comments for the Author):

The manuscript provides interesting and meaningful results about the role of transcription factor Upc2 on azole resistance mechanisms in *Candida auris*, that 90-100 clinical isolates are fluconazole resistant.

The manuscript is well written and easy to read and interpret the results.

Minor comments:

insert the statistical test used in the captions.

Response: We have added a mention of the statistical test in the figure legend.

In the figures, try changing the arrows by other symbols that indicate the differences between the compared strains. It's confusing.

Response: We have substituted the arrows by brackets to clearly show which columns are compared (also indicated in the figure legend).

Re: Spectrum03526-23R1 (Upc2-mediated mechanisms of azole resistance in *Candida auris*)

Dear Prof. Frederic Lamoth:

Your manuscript has been accepted, and I am forwarding it to the ASM production staff for publication. Your paper will first be checked to make sure all elements meet the technical requirements. ASM staff will contact you if anything needs to be revised before copyediting and production can begin. Otherwise, you will be notified when your proofs are ready to be viewed.

Sincerely,
Dimitrios Kontoyiannis
Editor
Microbiology Spectrum